# Examining the Relationship between Pes Planus Degree, Balance and Jump Performances in Athletes

**DOI:** 10.3390/ijerph191811602

**Published:** 2022-09-15

**Authors:** Fatma Neşe Şahin, Levent Ceylan, Hamza Küçük, Tülay Ceylan, Gökhan Arıkan, Sevcan Yiğit, Derya Çetin Sarşık, Özkan Güler

**Affiliations:** 1Department of Coaching Education, Faculty of Sport Science, Ankara University, Ankara 06830, Türkiye; 2Department of Coaching Education, Faculty of Sports Sciences, Sivas Cumhuriyet University, Sivas 58140, Türkiye; 3Yasar Doğu Faculty of Sport Sciences, Ondokuz Mayıs University, Samsun 55270, Türkiye; 4Department of Physical Education and Sport, Institute of Health Science, Sivas Cumhuriyet University, Sivas 58140, Türkiye; 5Mehmet Arabacı School of Physical Education and Sport, Harran University, Şanlıurfa 63000, Türkiye

**Keywords:** balance, pes planus, vertical jump, injury risk

## Abstract

The foot absorbs shocks with its arches, muscles, ligaments and joints, and bodyweight transmission and it pushes the body forward during all movement patterns. Pes planus is more important in sports activities that include balance and sports performance, such as walking, running, jumping, or transferring weight to a single lower limb. This study, conducted with this information in mind, aims to examine the effects of pes planus deformity on balance and vertical jump performance. Fifty athletes were included in the study. The presence of pes planus was evaluated according to the Feiss line. Balance measurements were performed with a Balance System SD Biodex. Vertical jump performance was recorded using an Omegawave jumping mat. The relationship between the pes planus grades of the participants and their balance and jump performances was analyzed using the Spearman correlation method. Vertical jump and Limit of stability (LOS) was significantly correlated with pes planus.

## 1. Introduction

Although the foot is seen as a static and rigid structure, it acts as a shock absorber. It plays an active role in locomotion and weight support structures. It has a flexible structure to balance the variations in the external environment [1,2]. The biomechanics of the system are complex, and many studies have analyzed the mechanism of transmitting ground reaction force [3,4,5]. The foot arches are fundamental for the dynamic function of the foot itself and during locomotion [6,7,8]. Foot arch types are generally neutral arch, pes planus (flat sole), and pes cavus (high arch) [9]. In a standard foot that adapts to the ground structure, contact with the ground occurs with the calcaneal tubercle, the first metatarsal head and the fifth metatarsal head. In the triangle consisting of these points, there are arches of the foot called the medial longitudinal arch (MLA), lateral longitudinal arch (LLA) and transverse arch. The MLA consists of the first metatarsal, medial cuneiform, navicular, talus, and calcaneus and is a weight-bearing and shock-absorbing structure [9,10]. Problems in any of these structures influence the physiological biomechanics of the foot [11]. Pes planus is a general term used to describe the decrease in height or complete flattening of the MLA during loading [12]. Pes planus is divided into flexible and rigid types [13,14]. There is a collapse in the arc in the rigid type, whether weight is given or not, but in the flexible type, the arc is lost during weight-bearing, while collapse is not observed in the unweighted condition [15,16]. Flexibile pes planus can be seen due to tibialis posterior dysfunction, malformations in the foot bones, loosening of the ligaments, shortening of the Achilles tendon, and loss of strength in the foot muscles [9]. Furthermore, in 1998, Otman et al. showed that flat feet require more energy expenditure due to their instability and increased muscle strain [17]. Pes planus causes pronation and plantar flexion in the foot and adduction of the talus and valgus of the calcaneus during weight bearing [9]. These changes in the normal alignment of the foot cause abnormal sensory input from the foot, which interferes with the proper muscle activity required for body posture and postural oscillations. The inability of the foot to perform its functions may cause problems in maintaining a balanced stance of the feet during walking, which is one of the most important activities of human beings, and this may cause the internal and external muscles to work harder to compensate for the deficiencies in the foot balance [18,19,20].

Balance is defined as the ability to maintain equilibrium by positioning our center of gravity over our base of support [21]. Balance is an important motor skill that affects athletic performance. Maintaining balance plays a crucial role in many sport movements such as running, jumping and kicking [21]. As balance is affected by physiological factors [21] (vestibular, visual and proprioception) [21] it can also be affected by musculoskeletal factors (pes planus, pes rectus, pes cavus). Pes planus is one of the most common musculoskeletal abnormalities in young people, which causes pain and difficulty during walking and running and jumping [22]. It has been found that athletes with pes planus exhibit poor ability to control foot movements in the ankle and foot complex, which may cause poor balance and jump performance [23]. However, one study found no significant correlation between the foot arch height and the vertical jump height in both normal-arched and pes planus individuals [24]. Other studies have shown that having pes planus did not affect motor performances in vertical jumps, sprints, and static balance [25]. There is still controversial literature on the impact of pes planus in athletic performance. Pes planus may not cause a decrease in athletic performance, but later complications of pes planus affect performance [26]. The current research, with this information in mind, aims to examine the effects of flexible pes planus deformity on balance and jump performance.

## 2. Materials and Methods

A total of fifty male team sport athletes (soccer 18, rugby 16, basketball 12 and volleyball, 4) volunteered to participate in this study. Characteristics of the participants were age: 18.6 ± 0.99 years; body weight: 67.52 ± 13.89 kg; height: 173.5 ± 9.21 cm; BMI: 22.22 ± 2.92 kg/m^2^. Athletes were excluded from the study if they had a lower extremity injury, vestibular disorder, or concussion within the previous six months. Participants were instructed not to perform exercises that may cause exhaustion 48 h before the tests and not to use stimulants such as alcohol, caffeine, or drugs in the 24 h before the study. This study was carried out in Ankara University Biomechanics Laboratory. The study was conducted according to the Declaration of Helsinki and was approved by the Ethics Committee of 19 Mayis University, Approval code 21-308, released in March 2021.

### 2.1. Evaluation of Pes Planus

The pes planus degrees of the participants were made by a single physiotherapist while standing on a hard floor with equal weight on both feet. In a normal foot, the tubercle of the navicular bone lies on the Feiss line drawn to the center of the medial malleolus and the metatarsophalangeal joint of the big toe. The degrees of pes planus are evaluated according to the departure of the navicular tubercle from this line and approaching the ground. If the tubercle of the navicular is reduced by 1/3 of the distance between the Feiss line and the ground, the pes planus is categorized as 1st degree, if it has decreased by 2/3 of it, it is classified as 2nd degree, and if it completely touches the ground, it is categorized as 3rd degree. If the tubercle of the navicular is above the Feiss line, it is defined as pes cavus [27,28,29,30]. The pes planus degrees of the athletes participating in the study were evaluated separately as right and left.

### 2.2. Assessment of Balance Ability

To measure postural stability, Limit of Stability (LOS) and postural stability index (PSI) this study used a commercially available balance device, the Biodex Balance System (BBS) (Biodex Medical Systems, Shirley, NY, USA), which consists of a movable balance platform that provides up to 20° of surface tilt in a 360° range of motion. The BSS has a circular platform that can freely move around the AP and medial-lateral (ML). The platform stability ranges from 1–12, with 1 representing the most significant instability. The familiarization protocol was implemented before the experiments. All participants were instructed to perform the balance test on five different days of the targeted weeks.

#### 2.2.1. Postural Stability Index

The main outcome of the PSI included the overall stability index (OSI), anterior-posterior stability index (APSI) and medial-lateral stability index (MLSI). The OSI represents the total variance of platform displacement (all directions), measured in degrees, with higher scores indicating worse postural control while the APSI and MLSI represent platform displacement in the sagittal and frontal planes respectively.

The test was performed with both feet and eyes open and subjects were allowed to see the real time feedback provided by the BSS computer interface. Athletes were asked to step on the platform of the BSS without footwear and assume a comfortable position while maintaining slight flexion in the knees. The postural stability test protocol consisted of 3 trials of 20 s of upright stance on both legs with 10 s of rest intervals between trials. A postural stability test was conducted on BBS with the platform set at level 12. When he could not maintain his balance during the measurement period when his hands or feet encountered the device, the measurement was canceled, and the test was repeated. A 60-s rest was given between all trials. The test was repeated 3 times and the best score was recorded for further analysis.

The intratester reliability of balance test procedure has been previously reported as ICC of the OSI was 0.43–0.82, and its intertester ICC was 0.42–0.70 [31].

#### 2.2.2. Limit of Stability Test

An individual limit of stability for standing balance is the maximum angle that the body can achieve from vertical without losing balance. The BBS provides scores for all eight directions as well as an overall score. Athletes were instructed to shift their weight to move a cursor toward each red, blinking target as displayed on the screen as quickly and with as much accuracy as possible. The nine targets were positioned in a circular fashion, which required the individual to shift their weight toward a target in the periphery and then return to a central location prior to shifting their weight to the next target in the various patterns defaulted on the screen. The manufacturer settings were used when administering the LOS; this resulted in 3 trials at each level with the average being used for analysis. The BBS provides scores for all 8 directions as well as an overall higher score indicated better performance. Two trials were completed for each athlete, and the results were recorded.

### 2.3. Assessment of Vertical Jump

Omegawave technology (Omegawave Ltd., Helsinki, Finland) was used to evaluate each athlete’s jumping performance. The vertical jump performances of the participants were tested with the Counter Movement Jump (CMJ) protocol [32]. The participants were asked to go out on the mat in light sports clothes (tights and t-shirt), make a quick downward squatting movement with their knees bent, from an upright position with both feet in contact with the ground, and then jump up with maximum force and land back on the mat. It was repeated five times. The highest jump height was recorded.

### 2.4. Statistical Analysis

Data analysis was performed using The Statistical Package for Social Sciences (v24.0, SPSS Inc., Chicago, IL, USA). Descriptive statistics were calculated for all tested variables. Descriptive statistics for all measure In addition Kolmogorov–Simirnov test was used to analyze the distribution of the variables It was determined that the data did not show normal distribution (*p* > 0.005). From this point of view, the Spearman rho test, which is a non-parametric comparison, was used in correlation analysis. Wilcoxon signed-rank test (one sample case) from the g-power 3.1.9.4 program was used to calculate the sample effect size in single groups While the sample size effect coefficient was calculated as 0.50 in the 96% (β) confidence interval, Critical t = 1.67811. df = 46.7464 in a sample size of 50 subjects. Statical significance set level was set at *p* = 0.05 for all comparisons.

## 3. Results

Fifty subjects were analyzed. The analysis performances of subjects are shown in Table 1.

In the limit of stability there is a correlation (spearman’s rank = 0.372) which is statically significant (*p* = 0.036) indicating that degree of pes planus increases as the degree of postural control decreases. In the vertical jump there is a correlation (spearman’s rank = 0.262) which is statistically significant (*p* = 0.048) indicating that as the degree of the pes planus increases the height of the vertical jump decreases. However, there are no significant correlations between pes planus and OSI, APSI, or MLSI. These results are described in Table 2.

## 4. Discussion

This study aimed to investigate the relationship of pes planus and balance ability and vertical jump in athletes. The correlation analysis between the vertical jump, balance and pes planus values showed a significant inverse correlation between the limit of stability vertical jump and right pes planus.

A 2016 study evaluated physical fitness parameters in men with and without pes planus. One hundred (*n* = 50 (degree of deformity 2 or more PP); *n* = 50 control group) healthy adult men participated in the study. As a result of the study, postural deviation and muscle shortness (in the gastrosoleus muscle group) were higher in the group with PP, while the time to stay in equilibrium was found to be lower. It was concluded that pes planus deformity negatively affects physical fitness in men [33]. A study conducted in 2018 examined the relationship between increased pronation in the hindfoot and physical performance. Sixty-four healthy young adults aged 18 to 25 years participated in the study. Hindfoot pronation was evaluated with the navicular drop test. The vertical jump, sidestep, and shuttle running tests were applied among the performance tests. It was concluded that increased pronation in the hindfoot decreased the sidestep test and shuttle running test [34]. In a study conducted in 2019, the effect of pes planus deformity on balance performance in athletes was investigated. Seventy-two athletes (38 men, 34 women) (*n* = 36 bilateral 3rd degree PP; *n* = 36 control group) were included in the study. According to the Feiss Line, Pes planus degrees and balance values were recorded with the HUBER 360 electronic device. It was concluded that the ability to balance on one foot on the dominant side of the athletes with pes planus on both feet was negatively affected. Researchers emphasized the importance of evaluating pes planus in branches where balance performance on the dominant extremity is important [35]. For pes planus grades by navicular drop test, static balance values were determined using the Flamingo balance test. In 2019, a static balance assessment was conducted for university students with and without pes planus. Twenty-one (*n* = 8 PP; *n* = 13 control group) female students were included in the study. As a result of the study, a significant difference was observed between pes planus and static balance. It was concluded that pes planus negatively affects the static balance. A study conducted in 2020 examined the relationship between navicular drop level and vertical jump performance. One hundred five healthy young people participated in the study. According to the results, no significant relationship was found between the navicular drop test and vertical jump measurement. Researchers have reported no relationship between longitudinal arc dynamics and physical performance such as vertical jump [36]. Three thousand four hundred ninety-three secondary school students participated in the study investigating the relationship between pes planus prevalence and obesity. Of the subjects, 31.2% were overweight and 21.2% of these obese participants were found to have flexible pes planus. While no significant relationship was found between gender and pes planus in all age groups, a significant relationship was found between obesity and pes planus [37]. In a study examining the relationship between foot arch structure and lower extremity injury, it was found that feet with excessive examples of pes cavus or pes planus are associated with the risk of various lower extremity injuries compared with those with a neutral arch [38].

## 5. Conclusions

The current study concluded that the balance and vertical jump performances of athletes with pes planus were negatively affected. In the presence of pes planus, we think that early evaluation is important to prevent the lower extremity alignment from being adversely affected and to minimize other problems that may cause it in the future. At this point, identifying those athletes with pes planus and applying the necessary corrective exercise approaches are important for the performance in sports.

## Figures and Tables

**Table 1 ijerph-19-11602-t001:** Descriptive statistics of all test variables.

Parameters	Minimum	Maximum	Mean	SD
Vertical Jump(cm)	28.1	57.3	43.5	6.54
LOS	36	80	57.91	11.26
OSI	0.1	1.8	0.4319	0.331
APSI	0.1	1.31	0.3	0.18
MLSI	0.01	0.9	0.21	0.17
Pes Planus (right)	1	2	1.69	0.46
Pes Planus (left)	1	2	1.67	0.47

LOS: Limit of stability; OSI: Overall stability index; APSI: Anterior-posterior stability index; MLSI: Medio-lateral stability index.

**Table 2 ijerph-19-11602-t002:** The correlation analyses of balance vertical jump and pes planus.

Parameters		OSI	APSI	MLSI	LOS	Vertical Jump
Right Pes Planus	Correlation Coefficient	−0.078	−0.010	−0.168	−0.372	−0.262
	*p*	0.606	0.950	0.265	0.036 *	0.048 *
Left Pes Planus	Correlation Coefficient	−0.066	0.047	−0.187	0.017	−0.105
	*p*	0.63	0.756	0.214	0.908	0.488

* *p* < 0.05; LOS: Limit of stability; OSI: Overall stability index; APSI: Anterior-posterior stability index; MLSI: Medio-lateral stability index.

## Data Availability

MDPI Research Data Policies.

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
