# Peer review of "Examining the Relationship between Pes Planus Degree, Balance and Jump Performances in Athletes"

_ijerph, 2022, doi:10.3390/ijerph191811602_

Round 1

Author Response

Dear reviewer,

Thank you for your valuable contribution and suggestion.

I will present evaluation of pes planus and jump performance with a figure.

All suggestions and corrections you mentioned have been made.

Thanks in advance

Sincerely

Reviewer 2 Report

Glad to review this paper. Below are my major concerns:

No info related to hypotheses development and testing.

Need to report more details regarding experiment subjects, such as competition level, sport programs, and injuries.

Sample size of 5 subjects is very limited.

Merely using Pearson correlation seems to be insufficient. The subject info (age, injury status, gender, etc.) would be covariates in the reported bivariate relationships. Authors are advised to considered more rigorous statistical method such as MANCOVA, ANCOVA, or structural equation modeling (if the sample size is sufficient large) with a bigger sample size.

Author Response

Dear reviewer,

I am glad to hear your suggestion.

We updated the statistical analysis as you specified

There are 50 participants in the study.

Sincerely,

Reviewer 3 Report

The autors report a study in which they investigate the effects of pes planus on motor performance.

The topic of the paper is suitable for the journal. It addresses an issue that is of relevance mainly to sports therapists and physiotherapists.

Introduction:

l. 65: "Pes planus is more influential in sports activities that involve balance and performance, such as walking, running, jumping, or transferring weight to a single lower extremity." It is important to address the extent to which pes planus influences sporting activities. Although it is stated at the end of the introduction that the pes plenus has a negative effect on sporting movements, there is no study evidence at this point.

Methods:

The subjects could be described more precisely (e.g. sporting activity).

It is described that only 1 physiotherapist judged whether pes planus was present or not. Was objectivity tested here once (> 1 observer). If so, was it concluded that the testing was objective and therefore only one observer was necessary? If this is not the case, this circumstance also influences the validity of the measurement to a significant extent. Measuring the pressure distribution of the feet would be another way to get objective information.

Counter Movement Jump: There is no justification why this test was chosen and not, for example, the Jump and Reach test. It is also unclear whether and how pauses were controlled.

Results:

table 1: The units of measurement are missing. In the present presentation, the data can only be interpreted with difficulty.
There is also no statement or data on how many people were diagnosed with pes planus.

table 2: Clarify more precisely which values represent the r-values and what the significances are.

Author Response

Dear reviewer,

I am glad to hear your suggestion.

We updated the statistical analysis as you specified

Sincerely

Round 2

Reviewer 1 Report

left and right pes planus should be frequency analysis. The other variables are descriptive statistics, but the statistical analysis has been reworked, table 1 is not modified. 

In statistical analysis, it should be explained in detail which statistical method was used.

Author Response

Dear Reviewer

Thank you for giving us the opportunity to submit a revised draft of the manuscript. We are grateful to you for your insightful comments on my paper

  • Statistical analysis is explained in detail which statistical method was used.

Sincerely,

Reviewer 2 Report

Most of issues have been corrected and improved in this revision, whereas some issues such as lacking of hypothesis development still exist. I will open to the decisions of my peer reviewers and editors.

Author Response

Dear Reviewer

Thank you for giving us the opportunity to submit a revised draft of the manuscript. We are grateful to you for your insightful comments on my paper.

 We have made revisions to the issues I mentioned below

  • We revised the introduction part for focusing on the impact of pes planus on sports and physical activity.
  • We explained the subjects in detail.
  • G power analysis was conducted and we have revised the method section
  • The ICC score was not obtained by analyzing the data of this study. It belongs to the validity and reliability study of the Biodex Balance System in the literature. we have added the reference to the study
  • Biodex Balance System is fully described and explained what the score mean
  • We explained in the method section, that the subjects had never done the test with the bodex balance system before, so a familiarization protocol was implemented before the experiments.
  • When we made the statistics of the data again, the spearman correlation analysis method was used. But we omitted to fix this in the manuscript

Reviewer 3 Report

The authors have improved some of the criticisms in their manuscript, but there are still some serious flaws, especially in the methodology and presentation of results.

Introduction: There is still a lack of studies focusing on the impact of pes planus on sports and physical activity.

Methods:

- The numerical material is presented inconsistently. Please round to one decimal point and use a point instead of a comma for decimal numbers.

- The subjects are still not described in detail. Which athletes are they or which sports do they do? Was a power analysis calculated to determine the number of subjects?

- With regard to balance, ICC values are reported. The question arises here as to how many people evaluated the measurements at all and from how many the ICC was calculated! Furthermore, it is unclear how many runs were performed.

- Operationalisation using the Biodex Balance System is not fully described. What do the scores mean?

- Overall, it would be important to report the internal consistency of the balance and jump results.

- It also remains unclear whether and if so what experience the subjects already had with the test procedures. If the subjects know the tests, this greatly affects the internal validity!

- The implementation of the test sequence is also important. How were these carried out?

Statistics:

- The significance level is given as follows: p < .05.

- Why was pearson and not spearman used? Were the results normally distributed (tested e.g. by Kolmogorov-Smirnov test)?

Results:

- Table 1: The units are still missing.

- Table 2: Where are the p-values and where the correlations?

- Were there any exercise effects in the test series and possibly also in the exercise phases?

Author Response

Dear Reviewer

Thank you for giving us the opportunity to submit a revised draft of the manuscript. We are grateful to you for your insightful comments on my paper.

  • We revised the introduction part for focusing on the impact of pes planus on sports and physical activity.
  • We explained the subjects in detail.
  • G power analysis was conducted and we have revised the method section
  • The ICC score was not obtained by analyzing the data of this study. It belongs to the validity and reliability study of the Biodex Balance System in the literature. we have added the reference to the study
  • Biodex Balance System is fully described and explained what the score mean
  • We explained in the method section, that the subjects had never done the test with the bodex balance system before, so a familiarization protocol was implemented before the experiments.
  • When we made the statistics of the data again, the spearman correlation analysis method was used. But we omitted to fix this in the manuscript
  • There is only one unit for vertical jump in the table. This is how other data comes out of the biodex balance system. In many articles in the literature, in which the biodex system is used for balance measurement, there is no unit besides the test data.
  • P valua is 0.05 and there are corrlation between right pes planus and vertical jump and Limit of stability score